# α-Pinene Enhances the Anticancer Activity of Natural Killer Cells via ERK/AKT Pathway

**DOI:** 10.3390/ijms22020656

**Published:** 2021-01-11

**Authors:** Hantae Jo, Byungsun Cha, Haneul Kim, Sofia Brito, Byeong Mun Kwak, Sung Tae Kim, Bum-Ho Bin, Mi-Gi Lee

**Affiliations:** 1Department of Applied Biotechnology, Ajou University, Suwon 16499, Korea; jesuswh1@gmail.com (H.J.); sofia@ajou.ac.kr (S.B.); 2Department of Biological Sciences, Ajou University, Suwon 16499, Korea; sunlits@ajou.ac.kr (B.C.); hanuel9695@ajou.ac.kr (H.K.); 3Department of Meridian and Acupoint, College of Korean Medicine, Semyung University, Chungbuk 27136, Korea; bmkwak@semyung.ac.kr; 4School of Cosmetic Science and Beauty Biotechnology, Semyung University, 65 Semyung-ro, Jecheon-si, Chungcheongbuk-do 27136, Korea; 5Department of Pharmaceutical Engineering, Inje University, Gimhae-si, Gyeongsangnam-do 50834, Korea; 6Department of Nanoscience and Engineering, Inje University, Gimhae-si, Gyeongsangnam-do 50834, Korea; 7Bio-Center, Gyeonggido Business and Science Accelerator, Suwon 16229, Korea

**Keywords:** natural killer cell, phytoncide, α-pinene, NK cell cytotoxicity, anticancer effect, ERK/AKT pathway

## Abstract

Natural killer (NK) cells are lymphocytes that can directly destroy cancer cells. When NK cells are activated, CD56 and CD107a markers are able to recognize cancer cells and release perforin and granzyme B proteins that induce apoptosis in the targeted cells. In this study, we focused on the role of phytoncides in activating NK cells and promoting anticancer effects. We tested the effects of several phytoncide compounds on NK-92mi cells and demonstrated that α-pinene treatment exhibited higher anticancer effects, as observed by the increased levels of perforin, granzyme B, CD56 and CD107a. Furthermore, α-pinene treatment in NK-92mi cells increased NK cell cytotoxicity in two different cell lines, and immunoblot assays revealed that the ERK/AKT pathway is involved in NK cell cytotoxicity in response to phytoncides. Furthermore, CT-26 colon cancer cells were allografted subcutaneously into BALB/c mice, and α-pinene treatment then inhibited allografted tumor growth. Our findings demonstrate that α-pinene activates NK cells and increases NK cell cytotoxicity, suggesting it is a potential compound for cancer immunotherapy.

## 1. Introduction

Natural killer (NK) cells are lymphocytes with a cytotoxic role, possessing an important function in immunity, as they are able to recognize and eliminate virus-infected (or “stressed”) cells, as well as cancer cells [1]. NK cells comprise 10% of the lymphocytes contained in the blood and peripheral lymphoid organs [2], containing abundant cytoplasmic granules and characteristic cell surface markers [3]. When NK cells are activated, they release proteins contained in their granules, such as perforins and granzymes, to attack invasive cells and induce apoptosis. NK cells and macrophages feed on microorganisms and produce interleukin 12 (IL-12), which, in turn, activates NK cells to secrete interferon-γ (IFN-γ), and IFN-γ subsequently activates macrophages to eliminate target cells [4].

Cancer immunotherapy is a type of strategy for cancer treatment that uses the individual’s natural immune system to combat cancer [5]. Moreover, many cancer patients who receive cancer immunotherapy treatment have shown a good prognosis [6,7,8]. Chimeric antigen receptor (CAR)-engineered T (CAR-T) cells are a known treatment with noticeable efficacy in cancer immunotherapy, allowing tumor selectivity and elimination by separating T-cells from the patient’s white blood cells and adding the specific CAR to target particular cancer cell antigens. Recently, the use of CAR-NK cells has been the subject of growing interest as a suitable candidate for anticancer research since, contrarily to CAR-T therapy, it induces minimal cytokine release and neurotoxicity, possesses multiple mechanisms for activating cytotoxic activity and has a high feasibility [9]. In fact, NK cells have revealed several new principles for recognizing and responding to tumors, demonstrating the potential positive effect on the clinical application of NK cells in cancer immunotherapy [10]. One example is the transfer of expanded and activated NK cells, in which autologous NK cells or the development of the CAR on NK cells by genetic engineering techniques facilitates adoptive transfer to patients. The adoptive transfer of NK cells has the advantage of being highly effective in promoting antitumor activity [11,12,13]. A second example is cytokine-based therapy. IL-2, IL-15 and IL-21 have been used to boost NK cell and T cell activity [14,15,16,17]. A third example is the targeting of the inhibitory receptors of NK cells, such as PD1/PD-L1 blockade [18,19], mAbs to KIRs [20,21], mAbs to NKG2A [22], TIGIT/CD96 blockade [23] and TIM-3 blockade [24,25]. These receptor modulations boost NK cell activity against tumor cells and metastasis [26].

It has been well documented that “forest bathing” enhances human innate immune cell activity, especially in NK cells, by boosting the expression of anticancer proteins [27]. This beneficial effect of “forest bathing” is known to be mainly due to phytoncides, aromatic volatile substances derived from trees. Phytoncides are most commonly released from conifers, especially pine and nut pine trees. Additionally, pine tree essential oil extracted from the leaf was found to contain 50 different terpene components, the majority being α- and β-pinene, myrcene, β-thujene and bornyl acetate. Among them, the content of α-pinene was revealed to be the highest [28].

Several types of terpene anticancer studies have been previously conducted. In particular, it was found that 37 types of monoterpenes present in the essential oils of plants possess anticancer effects [29]. Among several monoterpenes, anticancer studies of α-pinene have been established in various types of cancer and in vivo experiments, such as human hepatocellular carcinoma [30,31] and prostate cancer [32] in xenograft mouse models. Moreover, α-pinene is known to induce apoptosis by G2/M phage cycle arrest in human ovarian cancer and hepatocellular carcinoma [33,34]. In addition, α-pinene can inhibit the expression of matrix metalloproteinase-9 in human breast cancer cells, thereby inhibiting cancer invasion [35].

Phytoncides significantly increase NK cell activity with a concomitant increase in the expression levels of perforin and granzyme B, crucial effector molecules in NK cell-mediated cytotoxicity [36,37,38]. Furthermore, phytoncides have been observed to increase the number of CD56 molecules, which, along with CD107a, mark the activation of NK cells [39]. Lupeol, a triterpene, has been speculated to increase IFN-γ and CD107a via the activation of PI3K/AKT and Wnt/β-catenin signaling pathways [40]. In our previous study, the scent of phytoncide was revealed to have anticancer effects as well [41].

In this work, we studied the effects of phytoncide on NK cell activation. First, as phytoncide contains several compounds, we analyzed the effects of specific substances contained in phytoncides to test their efficacy in NK cell activation. Furthermore, by using α-pinene, known for its high effectiveness in NK cell activation, we analyzed NK cell cytotoxicity in mouse tumor allograft models and human NK-92mi cell lines. Additionally, since several reports demonstrate that NK cell activation occurs through the ERK/AKT signal pathway [42,43], an immunoblot assay was performed to confirm whether this was also the case for NK cells activated by α-pinene. Our results show that α-pinene increases the anticancer effect by accelerating NK cell activation and cytotoxicity via ERK/AKT signal pathways.

## 2. Results

### 2.1. Phytoncides Elevate the Expression of Activation Proteins in NK-92mi Cells

Phytoncide is a mixture of volatile components. Human NK-92mi cells were treated with each phytoncide component to determine which components of phytoncide activate NK cells. Perforin is a representative protein marker of the cytotoxicity of NK cells [44]. Therefore, we tested the effect of phytoncide constituents on the expression of perforin in NK-92mi cells after phytoncide treatment (Figure 1a). The results showed that α-pinene, o-cymene and camphor increased perforin mRNA expression in NK-92mi cells compared to the control (phorbol 12-myristate 13-acetate (PMA) and ionomycin treated). The mRNA expression of granzyme B [45], CD56 [46] and CD107a [47] also increased compared to the control (Figure 1b–d).

### 2.2. α-Pinene Enhances NK-92mi Cell Cytotoxicity

Based on our previous observation that α-pinene stimulates the expression of several NK cell markers (Figure 1a–d), NK cytotoxicity assays were performed to observe whether α-pinene activates NK-92mi cell cytotoxicity. The results indicated that α-pinene elevated NK-92mi cell cytotoxicity when NK-92mi cells were cocultured with target Hec-1A cells. Its cytotoxicity was increased 5.8-fold compared to the control and 1.8-fold compared to the PMA and ionomycin treatment (Figure 1e). In addition, when cocultured with K562 cells, the cytotoxicity of NK-92mi cells was increased further (by 1.7-fold) compared to the control and increased 1.2-fold compared to the PMA and ionomycin treatment (Figure 1f). This result demonstrates that HEC-1A cells are more sensitive to NK cytotoxicity than K562 cells, which is in accordance with a previous report demonstrating that the sensitivity of NK cytotoxicity depends on the target cell or its conditions [48]. Moreover, PMA and ionomycin treatment acting as an NK activator was found to be more cytotoxic than the control in HEC-1A cells. These results show that, even after NK activator treatment, α-pinene further increased NK cell cytotoxicity.

### 2.3. α-Pinene Induces NK Cell Cytotoxicity via the ERK and AKT Pathways

To determine whether the ERK/AKT pathway is involved in NK cell cytotoxicity in response to phytoncides [49], we performed immunoblot assays. α-Pinene treatment induced the activation of several ERK/AKT signaling molecules such as total ERK1/2, phospho-ERK1/2 (Thr202/Tyr204), AKT and phospho-AKT (Y312). The experiment was divided into a PMA and ionomycin-treated group and a control group. Cells were harvested at 5, 15, 30, 60, 120 and 240 min after α-pinene treatment to confirm the ERK and AKT pathway expression patterns over time. The emergence of ERK phosphorylation and AKT phosphorylation began around 5 min after α-pinene treatment (Figure 2). A similar time frame was manifested in PMA and ionomycin-activated NK-92mi cells. In the PMA and ionomycin treatment, the emergence of ERK phosphorylation and AKT phosphorylation occurred 5 min after treatment (Figure 2). Furthermore, to further confirm that α-pinene increases the ERK/AKT signaling pathway in NK cells, we conducted an immunoblot assay using mouse splenic NK cells. As a result, similarly to NK-92mi cells, it was found that α-pinene causes an increase in AKT phosphorylation in mouse splenic cells (Appendix A). However, α-pinene did not induce ERK phosphorylation, implying the presence of a cell- or model-type dependence on the ERK/AKT pathway for the response. α-Pinene enhanced the expression of signaling molecules in ERK and AKT pathway signaling, indicating that α-pinene activates NK cells through ERK and AKT signaling. In addition, we performed NK cytotoxicity toward K562 using α-pinene in combination with ERK and AKT inhibitors. As a result, it was confirmed that the NK cytotoxicity increased by α-pinene was reduced by ERK and AKT inhibitors (Appendix A).

### 2.4. α-Pinene Inhibited the Growth of CT-26 Colon Cancer Allografts in BALB/c Mice

CT-26 colon cancer cells were allografted in BALB/c mice by subcutaneously inoculating cells to the left flank to ensure that phytoncides would prevent cancer growth. The growth of the tumor allografts was inhibited significantly by α-pinene (Figure 3a–c). After α-pinene treatment was completed, the tumor-bearing mice were sacrificed and the tumors were excised to measure their weight. Compared to the control, tumor weight in mice treated with 40 mg/kg α-pinene was significantly reduced (Figure 3c). α-Pinene inhibited the growth of CT-26 allograft in BALB/c mice, and the inhibition ratio (%) compared to the control group was 42.83% in the 40 mg/kg α-pinene treatment group. These data indicate that α-pinene has an inhibitory effect on tumor growth.

### 2.5. α-Pinene Enhances Mouse Splenic NK Cell Cytotoxicity

To examine whether the NK cells were activated by phytoncides, we performed an NK cytotoxicity assay. The tumor-bearing mice were treated with 20 or 40 mg/kg α-pinene for 16 days. The mouse spleen was removed to isolate NK cells (effector cells). To measure the cytotoxicity of effector NK cells, the amount of calcein-AM released from target YAC-1 cells was measured. In response to the 40 mg/kg α-pinene treatment of control mice, NK cell activities were increased 1.8-fold (Figure 3d).

## 3. Discussion

In this study, we demonstrate that α-pinene, a terpene originating from phytoncide, enables the ERK and AKT phosphorylation signaling pathways, leading to NK cell activation—observed by increased expressions of CD56 and CD107a (Figure 4). NK cell activation further induces the release of perforins, which attach to the cancer cell membrane and open pores on its surface, allowing the diffusion of granzyme B proteins and inducing cell apoptosis.

Terpenes are a significant component of phytoncides and have many beneficial effects on the body’s biological activity. Forest environments are known to be rich in various terpenes, which are widely used for their anti-inflammatory, antitumorigenic or neuroprotective activities [50]. Many types of monoterpenes are known to have anticancer effects [29].

α-Pinene, one type of monoterpene, has been previously studied in various cancer cases. Most studies have been carried out with the monoterpene being directly delivered into cells or tumors transplanted in mice. However, in our experiment, cancer cells and tumor allografts were not directly treated with α-pinene. Instead, we measured immune cell activity and observed that human and mouse splenic NK cell cytotoxicity increased with α-pinene treatment (Figure 1b,c and Figure 3d). These results indicate that α-pinene has an indirect anticancer effect by boosting NK cell cytotoxicity.

CD56 and CD107a are markers associated with the activation and cytotoxicity in NK cells [47,51]. Our study demonstrates that α-pinene enables the ERK and AKT phosphorylation signaling pathways, thus leading to NK cell activation, as verified by the increased expression of receptors CD56 and CD107a. CD56 and CD107a are correlated with various forms of cytokine secretion production, including perforin, granzyme B and IFN-γ [52,53,54]. In a previous study, it was found that NK cells require PI3K-ERK1/2-dependent signaling activation for the killing of *Cryptococcus neoformans* [55]. Furthermore, Kwon et al. have demonstrated that phosphorylation of NF-κB activates NK cells via synergistic ERK and PI3K-Akt pathways [49]. Similarly, our results also indicate that α-pinene increases phosphorylation of the ERK/AKT pathway, thus inducing NK cell activation (Figure 2).

The stimulation of NK cells is controlled by a balance of inhibitory and activator receptors expressed by NK cells and target cells [56,57]. Interestingly, the results of this study showed different responses to NK cytotoxicity in Hec-1A and K562 cells (Figure 1e,f). Human NK-92mi cell cytotoxicity was 2.5% towards Hec-1A cells and 32% towards K562 cells, suggesting that the immune response was different for different tumor receptor types [58]. However, upon α-pinene treatment, NK cytotoxicity increased regardless of the type of cancer. These results suggest that α-pinene enhances NK cytotoxicity regardless of tumor receptor type.

Furthermore, one important consideration for further research of this topic, which is relevant for anticancer technology, is the mechanism by which phytoncides enter the body and reach NK cells. We hypothesize that phytoncides can dissolve in the blood when inhaled into the lungs. Afterward, dissolved phytoncide components can directly stimulate NK cell activation. According to our results, this hypothesis is plausible since α-pinene directly stimulates NK cells in a similar manner. Consequently, further research must be done to demonstrate this in vivo, particularly by measuring the concentration of phytoncide in the blood. Secondly, we hypothesized that phytoncides are able to stimulate the olfactory epithelium and regulate hormones. Among these hormones, vasoactive intestinal peptide (VIP) is known to be able to control NK cell activation [59,60].

In conclusion, the results of this study show an indirect anticancer effect via NK cell activation by phytoncides, suggesting that immunity increases when humans are frequently exposed to plants that release these compounds. Furthermore, our findings also reveal the potential use of α-pinene as a new approach in cancer immunotherapy by NK cell therapy, or even as a complement for existing cancer treatments.

## 4. Materials and Methods

### 4.1. Cell Culture and Reagents

NK-92mi cells (a human IL-2 independent NK cell line) HEC-1A cells (a human endometrial carcinoma cell line), K562 cells (a human lymphoblast cell line), CT-56 cells (a mouse colon carcinoma cell line) and YAC-1 cells (a mouse lymphoma cell line) were obtained from the American Type Culture Collection (Manassas, VA, USA). NK-92mi cells were cultured in α-MEM media supplemented with 100 U/mL penicillin, 100 U/mL streptomycin, 20% FBS (Hyclone, Logan, UT, USA), 1% vitamin solution X100 (Gibco, Waltham, MA, USA) and 2-mercaptoethanol (Gibco). HEC-1A, K562, CT-26 and YAC-1 cells were cultured at 37 °C in DMEM/F12 media supplemented with 100 U/mL penicillin, 100 U/mL streptomycin and 10% FBS in a humidified incubator in the presence of 5% CO_2_. All other chemicals were purchased from Sigma-Aldrich (St. Louis, MO, USA) unless otherwise specified.

### 4.2. Mice Tumor Allografts

All procedures, including mice handling, were executed at the Department of Meridian and Acupoint of the College of Korean Medicine at Semyung University (smecae 20-07-01, approved on 1 July 2020), according to the guidelines approved by the Institutional Animal Care and Use Committee (IACUC). Female BALB/c mice, 6 weeks of age, were obtained from Joongah Bio (Suwon, Korea). Animals were housed in sterile plastic cages at controlled temperature (26 ± 2 °C), lighting (12 h) and humidity (60 ± 5%). Food and water were available ad libitum. To generate tumor allografts, CT-26 cells were subcutaneously injected at 1 × 10^7^ cells/200 μL in saline into the dorsal flank of female BALB/c mice (*n* = 10 mice/group), as described previously [61]. After being allowed to grow for 2 weeks to reach the proliferative phase, mice were subdivided into three groups for α-pinene (Sigma-Aldrich) treatments.

### 4.3. Tumor Volume Estimation

The sizes of the allografted tumors were measured externally every 2 days from the last measured date, and the tumor volumes were obtained according to the equation Volume = (L × S^2^)/2, where L and S are the lengths of the major and minor axes, respectively. The animals were monitored for 30 days after allograft implantation.

### 4.4. Mouse Splenic Natural Killer Cell Cytotoxicity

The mouse spleens were isolated, and red blood cells were removed as described previously [62]. The spleens were mashed through a 100-µm cell strainer (Corning Inc., Corning, NY, USA) in complete media consisting of DMEM/F12 media (Gibco) with 10% fetal bovine serum and antibiotics as described above. Spleen mononuclear cells were obtained by centrifuging the mashed spleen cell suspension on 4 mL of Histopaque-1077 (Sigma). The cells in the interface were withdrawn and washed three times with complete media. The resulting mononuclear cells were resuspended in complete media at 1 × 10^6^ cells/mL. YAC-1 cells, used as the target cells of spleen NK cells, were loaded with calcein-AM (Sigma), according to the method of Roden et al. [63]. The loading of YAC-1 cells (1 × 10^6^ cells/mL) was performed at a final calcein-AM concentration of 10 μM for 30 min at 37 °C. Purified mononuclear effector cells and labeled YAC-1 target cells were added at a ratio of 20:1 to a 96-well plate and cocultured for 4 h at 37 °C. Following centrifugation at 2000 rpm for 3 min, 150 μL of the supernatant from each well was collected to measure the fluorescence of the calcein released into the media using a microplate reader. The absorbance of the contents of each well was measured at the excitation and emission wavelengths of 485 and 538 nm, respectively. Spontaneous calcein release was determined by culturing the labeled YAC-1 cells in complete media without effector NK cells. Total fluorescence was acquired from wells where target cells were incubated with a lysis buffer consisting of 50 mM sodium borate and 0.1% Triton X-100 (pH 9.0). Specific lysis was calculated according to the following equation:(1)$$\%\;\mathrm{cytotoxicity}=\left[\;1-\frac{\mathrm{experimental}\;\mathrm{fluorescence}-\mathrm{spontaneous}\;\mathrm{fluorescence}}{\mathrm{total}\;\mathrm{fluorescence}-\mathrm{spontaneous}\;\mathrm{fluorescence}}\right]\times 100$$

### 4.5. Natural Killer Cell NK-92mi Activation

NK-92mi cells cultured at 1 × 10^7^ cells/mL were activated by PMA at 250 ng/mL and ionomycin at 1 μM for 6 h. Immediately after activation by PMA and ionomycin, the NK-92mi cells were further treated with phytoncide compounds (namely α-pinene, terpinolene, o-cymene and camphor) at 100 μM each in the cell culture media for 48 h before being subjected to reverse transcription PCR, real-time PCR, immunoblot and NK cell cytotoxicity assay.

### 4.6. Reverse Transcription and Real-Time PCR

Trizol reagent (Life Technologies, Waltham, MA, USA) was used for the extraction of total mRNA, according to the manufacturer’s instructions. Approximately 1 μg of each total RNA was subjected to reverse transcription reaction at 37 °C for 1 h using AccuPower RT premix (Bioneer, Daejeon, Korea) as per the manufacturer’s instructions. PCR was performed using a Mastercycler (Eppendorf, Hamburg, Germany). PCR was carried out for each sample with appropriate PCR primer sets using AccuPower PCR PreMix (Bioneer). The primers for perforin (5′-GTTTCCATGTGGTACACACTC-3′; 5′-GTGCCGTAGTTGGAGATAAG-3′), granzyme B (5′-TTCCTGATACGAGACGACTT-3′; 5′-TTTCACAGGGATAAACTGCT-3′), CD56 (5′-GATTCATAGTCCTGTCCAACA-3′; 5′-GACCTGAATGTCCTTGAAGTT-3′), CD107a (5′-ACACTCACTCTCAATTTCACG-3′; 5′-CTGCCCTGATGTCAGTTATAG-3′) and human GAPDH (5′-AACCTGCCAAATATGATGAC-3′; 5′-TTGAAGTCAGAGGAGACCAC-3′) (as a loading control) were purchased from Bionics (Seoul, Korea). Each primer sequence was designed based on the full-length mRNA sequence using Primer3 software (http://bioinfo.ut.ee/primer3-0.4.0/primer3/). Real-time PCR gene expression analyses were performed using CYBR Q Green 2× qPCR Master Mix (CellSafe, Yongin, Korea). Real-time PCR was performed with an Agilent AriaMx Real-Time PCR System (Agilent, Santa Clara, CA, USA).

### 4.7. NK Cell Cytotoxicity Assay

Two target cell lines, namely K562 cells and HEC-1A cells, were loaded with calcein-AM at 10 μM in media for 30 min at 37 °C and subsequently washed with phosphate-buffered saline three times. Stained target cells were incubated with NK-92mi cells as effector cells at a ratio of 20:1 for 4 h at 37 °C. After incubation, the cell plate was centrifuged at 2000 rpm for 10 min. The supernatant (150 μL) from each well was collected to measure calcein-AM fluorescence released into the media using a microplate reader (SpectraMAX GeminiXS). The absorbance was measured at excitation and emission wavelengths of 485 and 538 nm, respectively. Based on fluorescence intensity, NK cell cytotoxicity was calculated using Equation (1).

### 4.8. Immunoblot Assay

NK-92mi cell lines were seeded in 75T cell culture flasks at 1 × 10^7^ cells/mL. Treatment with 100 μM α-pinene was carried out to activate the NK-92mi cells. After treatment with α-pinene for 48 h, immunoblotting was performed. Briefly, NK-92mi cells were harvested by centrifugation at 2000 rpm for 3 min, and the cell pellet was lysed with RIPA buffer (cell lysis buffer). Bradford assay was used to determine the protein concentration. For each sample, 20 μg of protein was loaded onto a 15% polyacrylamide gel and subjected to SDS-PAGE. The proteins on the polyacrylamide gel were transferred to a PVDF membrane (Roche, Basel, Switzerland) and incubated for 2 h at room temperature with primary antibodies against ERK1/2, phospho-ERK1/2 (Thr 202/Tyr204), AKT, phospho-AKT (Y312) or β-actin. The membranes were washed further with TBS-T three times and incubated with secondary antibodies for 1 h at room temperature. All antibodies were purchased from Cell Signaling Technology (Danvers, MA, USA). Immunoblot bands were detected using an enhanced chemiluminescent substrate (Thermo Scientific, Waltham, MA, USA) and exposed to a luminescent image analyzer LAS-3000 (Fuji, Tokyo, Japan). Las-3000 Image Reader software was used to detect the dots on the membrane.

### 4.9. Statistical Analysis

A two-tailed Student’s *t*-test was used for statistical analysis between two groups.

## Figures and Tables

**Figure 1 ijms-22-00656-f001:**
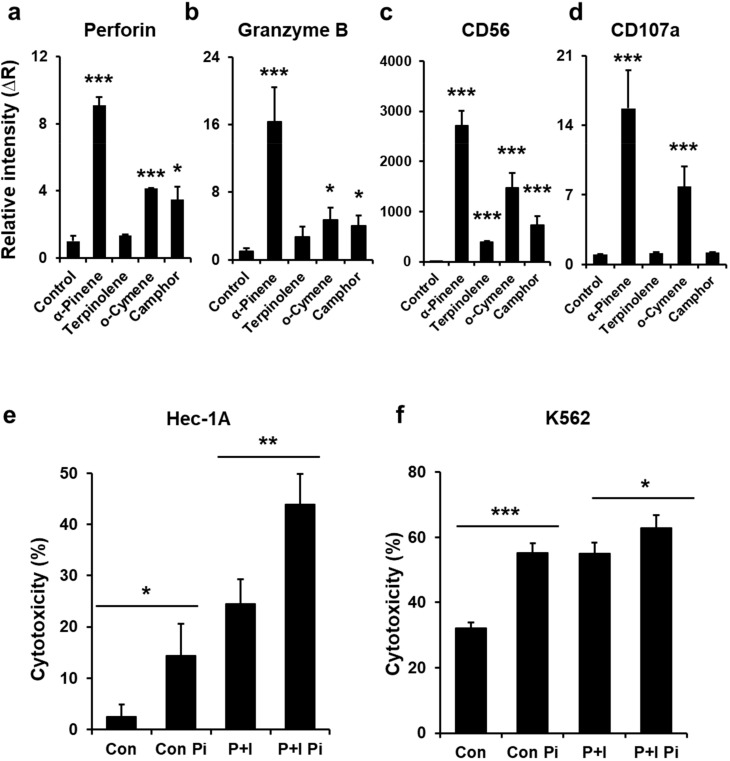
Phytoncides increase natural killer (NK) cell activation and cytotoxicity in NK-92mi cells. Human NK-92mi cells were activated by incubation with phorbol 12-myristate 13-acetate (PMA) and ionomycin for 6 h after treatment with α-pinene, terpinolene, o-cymene and camphor at 100 μM. The expression levels of (**a**) perforin, (**b**) CD56, (**c**) granzyme B and (**d**) CD107a mRNA were analyzed with real-time PCR. The data are expressed as the mean ± SD of three independent experiments (* *p* < 0.05, *** *p* < 0.005). Human NK-92mi cells were incubated with target K562 cells or HEC-1A cells, which were prelabeled with calcein-AM for 4 h, as described. Calcein-AM released from K562 and HEC-1A target cells was measured by a fluorescence microplate reader. (**e**) NK-92mi cytotoxicity towards Hec-1A cells. (**f**) NK-92mi cytotoxicity towards K562 cells. Con: control, Con Pi: control with α-pinene treatment, P+I: PMA and ionomycin activation, P+I Pi: PMA and ionomycin activation with α-pinene treatment. Results are expressed as the mean ± SD of three independent experiments (* *p* < 0.05, ** *p* < 0.01, *** *p* < 0.005).

**Figure 2 ijms-22-00656-f002:**
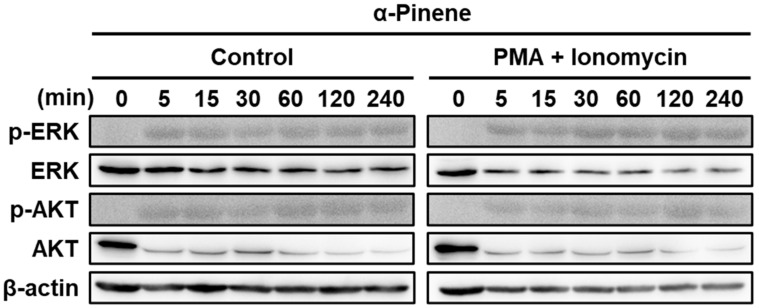
α-Pinene increases the expression of signaling molecules in the ERK/AKT signaling pathway in NK-92mi cells. Human NK-92mi cells were treated with α-pinene at 100 μM for 48 h. The total cell lysate was obtained and subjected to immunoblotting. Two groups were tested. NK-92mi control: NK-92mi cells without activation. PMA + ionomycin treatment: NK-92mi activated by PMA + ionomycin.

**Figure 3 ijms-22-00656-f003:**
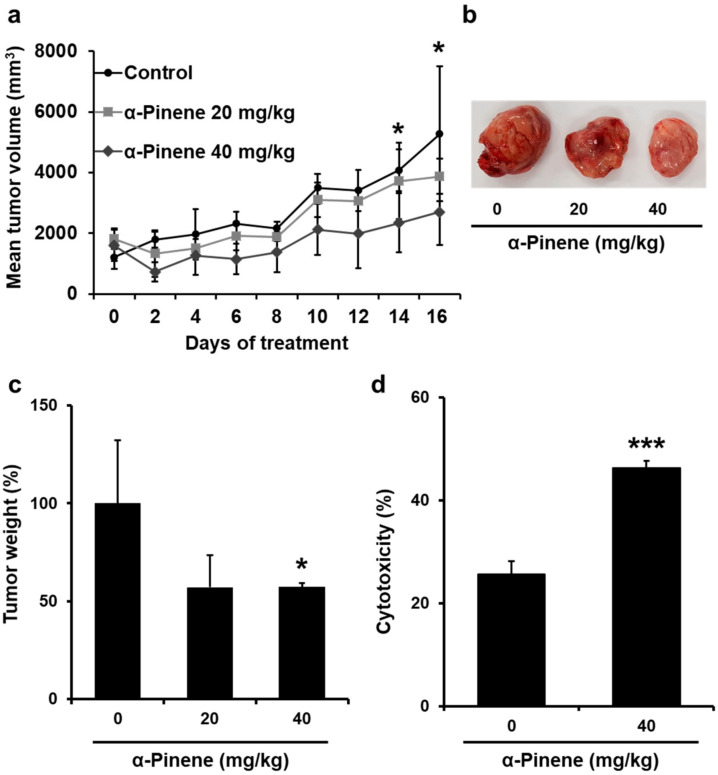
α-Pinene inhibited the growth of CT-26 colon cancer allografts in BALB/c mice by increasing NK cytotoxicity. (**a**–**c**) Tumor volume and weight were significantly decreased by α-pinene 40 mg/kg treatment compared to the control (* *p* < 0.05, *** *p* < 0.005). (**d**) The cytotoxicity of mouse splenic NK cells was measured. Splenic NK cells were isolated from those of normal or tumor-transplanted mice and incubated with YAC-1 target cells prelabeled with calcein-AM as described in Section 4. Hank’s Balanced Salt Solution was used for spontaneous fluorescence from YAC-1 cells without effector NK cells, and total fluorescence was acquired from wells where YAC-1 cells were incubated with a lysis buffer. NK cell cytotoxicity was calculated as described in Section 4.

**Figure 4 ijms-22-00656-f004:**
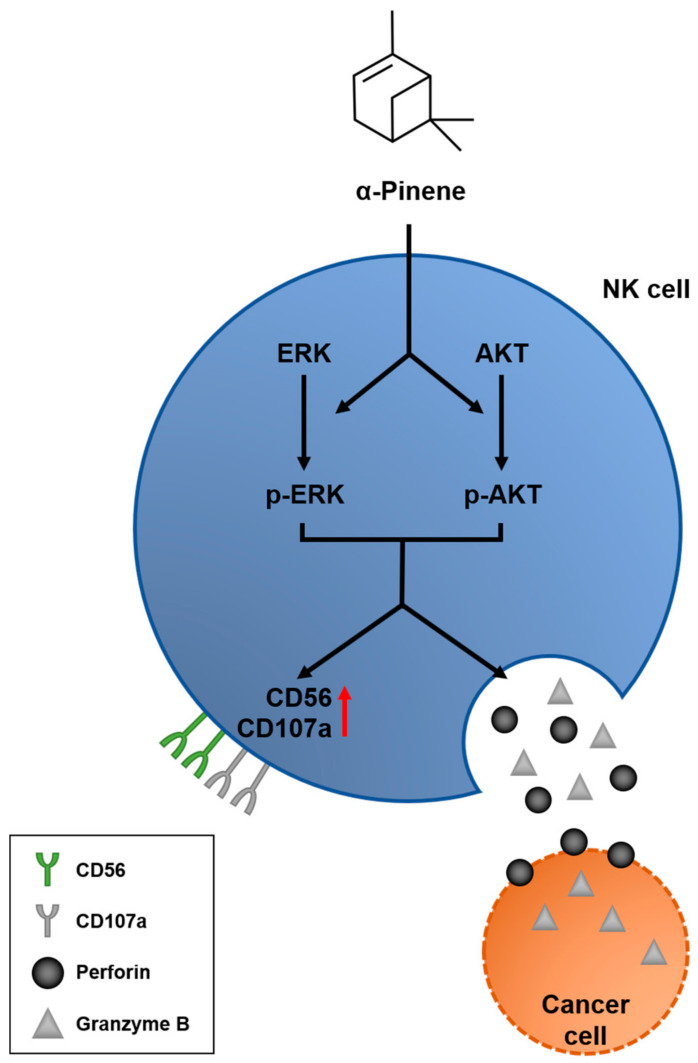
Schematic model of α-pinene promoting an anticancer effect via NK cell activation. α-Pinene stimulates ERK and AKT phosphorylation signaling pathways and leads to NK cell activation, which is confirmed by the increased expression of receptors CD56 and CD170. Consequently, the NK cell product proteins perforin and granzyme B are induced, leading to cancer cell death.

## Data Availability

Data is contained within the article text and figures.

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
