# Peer review of "α-Pinene Enhances the Anticancer Activity of Natural Killer Cells via ERK/AKT Pathway"

_ijms, 2021, doi:10.3390/ijms22020656_

Round 1

Reviewer 1 Report

The publication entitled: "α-Pinene enhances the anti-cancer activity of natural killer cells via ERK / AKT pathway" is a very good work worth publishing.
However, it needs to be significantly improved. For now, it contains many, not bugs, but terrible shortcomings.

Some of them are:
1. There is probably something that can be added to the keywords section, as the topic presented in the work is more extensive.
2. The admission is too poor and needs extensive expansion, there is a lot of work on the internet that can be referred to and the admission is greatly improved. It's too short for now. After typing a few keywords from the work, I found many good works that the authors did not even look at, or looked at but did not write about it.
3. Where is the purpose of the work, this is one sentence from the end of the introduction? After that, practically an explanation of what the authors got. so why the rest of the work. Please expand this paragraph.
4. Figure 1 - why is there a different color of the axle caption? And the bottom axis is somehow cut off - in the sense of its signature.
5. Figure 4 can be slightly improved. It's kind of slightly blurry, except in my version of the manuscript.
6. The discussion is terribly poor, for such a good magazine the authors probably got a little too little sense of the subject. Please refine the discussion considerably. Discussion is not another way (stylistics) of recording the results.
7. And where is the summary section? Does something from this research show or is it just that something influenced something? It is a pity that the authors want to sell such a cool idea anyway.

I recommend the work for publication after major correction.
I would like to see the changes in yellow in manuscript.

Author Response

Dear Reviewer, 

Attached is the revised manuscript according to reviewer's comments. We are resubmitting for consideration of publication in International Journal of Molecular Sciences (ijms-1039496). Here is the answer based on the reviewer 1 comments and suggestions. We also highlighted the revised sentences and paragraphs in the revised manuscript.

Sincerely,

Reviewer 2 Report

In the manuscript entitled "α-Pinene enhances the anti-cancer activity of natural killer cells via ERK/AKT pathway", the authors tried to determine how phytoncide affect the tumor cell growth. First the authors identified three constituentes of phytoncide (α-pinene, o-cymene, and camphor) can increase the expression of marker genes for the cytotoxicity of NK cells at the mRNA level. Following that, the authors further investigated the effect of pinene on human NK-92mi cells and CT-26 colon cancer in vitro and in vivo, and they claimed ERK/AKT pathway is involved in pinene enhanced cytotoxicity of NK cells. Regarding this manuscript, I have several concerns: 

1) The authors used several kinds of human cancer cells, including uterus endometrial cancer cell, lung cancer cell (in vitro) and  CT-26 colon cancer cell (in vivo). It is very confusing why the authors used different kinds of cancer cells for in vitro and in vivo studies. Why did not you consistently use one or two for both in vitro and in vivo? 

2) How does ERK/AKT change in NK cells in vivo when you treated mice with pinene in both control and tumor bearing mice?

3) Will it inhibit the cytotoxicity of NK cells when you treated with ERK/AKT inhibitors? or when you pre-treated NK cells with ERK/AKT inhibitors in vitro studies?

4) In Figure 2, the total ERK/AKT dereased a lot even when you treated pinene for 5 min but the phosphorylated ERK/AKT did not increase that much? how can you explain this?

5) Why did not I see ERK/AKT as key words in the Abstract and Discussion?

6) Line 87, check the writing.

Author Response

Dear Reviewer, 

Attached is the revised manuscript according to reviewer's comments. We are resubmitting for consideration of publication in International Journal of Molecular Sciences (ijms-1039496). Here is the answer based on the reviewer 2 comments and suggestions. We also highlighted the revised sentences and paragraphs in the revised manuscript.

Sincerely,

Round 2

Reviewer 1 Report

The authors responded very well to my comments. I believe that the work may already be published in this form.

Reviewer 2 Report

The authors performed further experiments and made more explainations. They have addressed my concerns.